# Clinical Applications of Fetal MRI in the Brain

**DOI:** 10.3390/diagnostics12030764

**Published:** 2022-03-21

**Authors:** Usha D. Nagaraj, Beth M. Kline-Fath

**Affiliations:** 1Department of Radiology and Medical Imaging, Cincinnati Children’s Hospital Medical Center, Cincinnati, OH 45229, USA; beth.kline-fath@cchmc.org; 2Department of Radiology, University of Cincinnati College of Medicine, Cincinnati, OH 45229, USA

**Keywords:** fetal MRI, brain, ventriculomegaly, cavum, posterior fossa

## Abstract

Fetal magnetic resonance imaging (MRI) has become a widely used tool in clinical practice, providing increased accuracy in prenatal diagnoses of congenital abnormalities of the brain, allowing for more accurate prenatal counseling, optimization of perinatal management, and in some cases fetal intervention. In this article, a brief description of how fetal ultrasound (US) and fetal MRI are used in clinical practice will be followed by an overview of the most common reasons for referral for fetal MRI of the brain, including ventriculomegaly, absence of the cavum septi pellucidi (CSP) and posterior fossa anomalies.

## 1. Introduction

Fetal magnetic resonance imaging (MRI) has become a readily available tool that can be used to evaluate the fetal brain. Possible or suspected fetal brain anomalies identified on routine screening fetal ultrasound (US) can be referred for fetal MRI in order to gain more information about the developing brain. In this review, the use of fetal ultrasound to screen for brain anomalies will be briefly discussed, followed by a short summary of fetal MRI use and exam protocol. The majority of this review will highlight and discuss utility of fetal MRI in the brain in the most common scenarios encountered in clinical practice: further evaluation of ventriculomegaly, absent cavum septi pellucidi (CSP) and posterior fossa abnormalities [1].

## 2. Imaging Modalities

### 2.1. Fetal Ultrasound

Fetal US is the screening tool of choice for congenital abnormalities of the brain, with a second-trimester US being the current standard of care in the United States [2,3]. It is believed that most congenital brain anomalies can be first detected on fetal US. One of the largest retrospective series found that up to 95% of fetal brain anomalies can be identified on routine screening US. This is accomplished by examining three main components: lateral ventricular size, presence of the CSP and cisterna magna size. Ventricular size is the most important component of this exam and can detect up to 88% of central nervous system (CNS) anomalies on its own [4].

However, fetal US does have some technical limitations as image quality can be compromised by maternal obesity, decreased amniotic fluid, fetal positioning and calvarial ossification. In addition, its relative lack of diagnostic specificity usually requires that the detection of a sonographic abnormality undergo further work up. Genetic testing and TORCH (Toxoplasmosis, Other (syphilis, varicella-zoster, parvovirus B19), Rubella, Cytomegalovirus and Herpes) infection screening are highly recommended in patients with ventriculomegaly on fetal US [5]. Follow-up third trimester US may also be considered, though with advancing gestational age, low fetal head positioning can limit the visualization of the intracranial structures as the maternal pelvic bones can narrow the acoustic window. Postnatal brain imaging may also be considered to follow-up suspected brain abnormalities identified on fetal US. Available imaging modalities include postnatal head US, computed tomography (CT) or MRI. However, fetal MRI provides the best evaluation of the fetal brain in vivo and can be instrumental in prenatal diagnosis of brain pathology.

### 2.2. Fetal MRI

Fetal MRI has become an effective method in the prenatal evaluation of suspected brain anomalies. It is now readily available in most centers in the United States and does not require maternal sedation [6]. Fetal MRI greatly improves diagnostic accuracy in brain pathology when compared to fetal US alone. This results in information that can change prognosis, prenatal counseling and prenatal/perinatal management [7,8]. When to perform a fetal MRI for suspected CNS abnormality during pregnancy is a frequently asked question and should be determined on a case by case basis. In general, a third-trimester fetal MRI is preferred because of overall increased fetal size and brain development allowing for more information to potentially be obtained [9]. However, there are situations in which a second trimester fetal MRI is preferred. If the patient wishes to consider termination of the pregnancy, fetal MRI would be best performed soon after the fetal US since termination is not legal after 24 weeks gestational age (GA) in most states. Fetal MRI might also be indicated in the second trimester if the patient is a potential candidate for fetal surgery, as in fetuses being evaluated for prenatal closure of an open spinal dysraphism, in which surgery is typically offered before 26 weeks of GA [10].

Fetal imaging protocols vary amongst institutions but should include dedicated imaging of the brain. National and international practice guidelines have been described with recommended imaging protocols [11]. At our center, a brain malformation protocol includes T2 single shot fast spin echo (T2-SSFSE) images through the brain at 3–4 mm-thick slices in three orthogonal planes, at least two stacks in each plane to the radiologist’s satisfaction. Cerebrospinal fluid (CSF) on this sequence is bright, and these images provide the best contrast resolution of the brain parenchyma based on the relatively high water content of the developing brain. We also employ balanced turbo-field echo/fast imaging employing steady-state acquisition (BTFE/FIESTA) images through the brain in three orthogonal planes for improved spatial resolution. This sequence is heavily T2-weighted, such that CSF is also bright highlighting water and craniospinal interfaces; however, differences in regional brain water content are not as pronounced as the T2-SSFSE images. We also use axial diffusion weighted imaging (DWI) to examine brain tissue cellularity and water movement, which can be useful in the evaluation of cerebral infarction or hemorrhage. We add echo planar imaging (EPI), which allows for increased sensitivity to intracranial hemorrhage [12]. Axial T1-weighted imaging of the brain is utilized to evaluate for the presence of intracranial lipoma, hemorrhage or thrombosis, which can be bright on T1. Smallest field of views possible are utilized (Figure 1).

Fetal MRI can be performed on a 1.5 or 3.0 Tesla magnet. While there are reports of improved image quality in certain scenarios on 3.0 Tesla, our preference in routine clinical practice is to use a 1.5 T due to decreased imaging artifacts and shorter scan times [13].

## 3. Ventriculomegaly

Ventriculomegaly of the lateral ventricles, defined as an atrial width of 10 mm or greater, is the most common brain anomaly identified on fetal ultrasound and is a common reason for referral for fetal MRI [12,13,14]. On ultrasound, the identification of ventriculomegaly is sensitive for the detection of other brain anomalies; however, fetal MRI is an excellent tool in improving specificity. Ventriculomegaly can either be the result of obstructive hydrocephalus or decreased cerebral volume, for which fetal MRI can be helpful in differentiating the two, understanding that the two can co-exist [15]. Additionally, of note is that isolated ventriculomegaly, particularly when mild, may be an incidental finding with normal neurologic outcomes [16,17]. In this section, we will briefly discuss some of the most common causes of obstructive hydrocephalus and decreased cerebral volume in the fetus, along with the role fetal MRI plays in prenatal diagnosis and management.

### 3.1. Obstructive Hydrocephalus

The term obstructive hydrocephalus denotes that there is evidence of increased intracranial pressure secondary to a point of obstruction within the ventricular system. Effacement of the extra-axial CSF spaces over the cerebral convexities in the setting of ventriculomegaly may be a potential clue in identifying obstructive hydrocephalus. The two most common causes of obstructive hydrocephalus in the fetus are aqueductal stenosis and Chiari II malformation [18,19].

#### 3.1.1. Aqueductal Stenosis

Aqueductal stenosis (AS) is the most common cause of obstructive hydrocephalus in the fetus, in which the level of ventricular obstruction is located within the cerebral aqueduct. AS can be caused by genetic abnormalities (LCAM1 mutation, dystroglycanopathy) or acquired conditions (hemorrhage, infection) [20,21,22]. Fetal MRI can help confirm the diagnosis by illustrating classic imaging findings of ventriculomegaly of the lateral and 3rd ventricles, 3rd ventricular funneling, and normal size 4th ventricle. Fetal MRI can also aid in identifying potential causes such as intraventricular hemorrhage or intracranial mass/cyst (Figure 2). Additionally, it is important to note that fetal MRI can identify additional associated brain anomalies such as brainstem abnormalities, vermian hypoplasia, diencephalic–mesencephalic junction dysplasia or rhombencephalosynapsis. These findings can be of particular importance during prenatal counseling and guiding prenatal and perinatal management [23]. An example of this is rhombencephalosynapsis, which is a condition in which there is incomplete separation of the cerebellar hemispheres with associated partial or complete absence of the cerebellar vermis, seen in up to half of patients with prenatally diagnosed AS. This diagnosis is known to be associated with increased feeding assistance and ventricular shunting in the neonatal period, important for prenatal counseling [24].

#### 3.1.2. Chiari II Malformation

Chiari II malformation is defined as the constellation of intracranial findings (mainly hindbrain herniation) in the setting of open spinal dysraphism, usually myelomeningocele, and is one of the most common causes of obstructive hydrocephalus in the fetus [25]. Fetal ultrasound is relatively sensitive and specific in the diagnosis. The two most frequently reported sonographic signs are the “lemon sign” marked by bifrontal concavity of the calvarium, seen in 98% of cases before 24 weeks GA, and the “banana sign” marked by appearance of the cerebellum wrapped around the brainstem [26,27]. While fetal MRI does not typically add to the specificity of diagnosis, fetal MRI in this patient population is of particular importance in the context of fetal surgery. About 10 years ago, the Management of Myelomeningocele (MOMS) trial established safety and efficacy in the prenatal repair of open spinal dysraphisms, such that fetal surgery is now considered an acceptable and in certain cases desirable alternative to postnatal closure [28,29]. Fetal MRI is essential in these patients in order to confirm and characterize hindbrain herniation, such that the best candidates for surgery may be selected. Fetal MRI also allows for the evaluation of the open neural tube defect, important for planning fetal surgery in this population [30,31] (Figure 3).

### 3.2. Decreased Cerebral Volume

Decreased cerebral volume can be acquired or congenital, and in some cases both. While isolated mild ventriculomegaly in the fetus is usually incidental, further investigation by follow-up imaging and/or laboratory investigations is warranted. Fetal MRI can help elucidate causes and the extent of intracranial abnormalities since ventriculomegaly on fetal US is sensitive, though not specific, for congenital brain anomalies. Acquired causes are the result of in utero injury (vascular, infection, trauma, genetic) to the developing brain. Fetal MRI can help define the degree of ventriculomegaly (mild, moderate, severe) and look for areas of parenchymal destruction which may be marked by porencephalic cysts or absence (focal or diffuse) of the brain parenchyma [32]. On the severe end of the spectrum, we can see diffuse cystic encephalomalacia of the brain in the setting severe hypoxic ischemic injury [33]. Another pattern of injury that can be confirmed on fetal MRI is hydranencephaly marked by near complete absence of the supratentorial brain parenchyma [34] (Figure 4). Malformations of cortical development including gray matter heterotopia, schizencephaly and polymicrogyria can also result in fetal ventriculomegaly and can be detected on fetal MRI; however, they are often better assessed on postnatal brain MRI [35]. While 2D biometric measurements can be helpful in assessing cerebral volume on fetal MRI, applications for fetal brain volumetric analysis hold promise for the future [36].

## 4. Absent Cavum Septi Pellucidi

Evaluation of the cavum septi pellucidi (CSP), more commonly referred to as the “cavum septum pellucidum,” is an established part of routine screening fetal US and its absence can be a marker of a number of abnormalities [37]. Fetal MRI can help to narrow the differential diagnosis in cases of absent CSP. Differential considerations include holoprosencephaly spectrum, corpus callosum anomalies, hypoplastic optic nerve syndrome or isolated septal deficiency. In this section we will describe the role of fetal MRI in diagnosing these entities [38].

### 4.1. Holoprosencephaly Spectrum

Holoprosencephaly refers to a group of brain anomalies characterized by abnormal differentiation and midline separation of the developing forebrain, resulting in abnormal continuation of cerebral structures across the midline [39]. The DeMyer classification is the most frequently referenced, in which holoprosencephaly is divided into alobar, semilobar and lobar subtypes, all described as lacking a septum pellucidum [40]. In clinical practice, holoprosencephaly is seen as a diverse spectrum of abnormalities in which areas of incomplete midline separation of varying, sometimes subtle, degrees can be seen in almost all aspects of the brain, even in cases with an intact septum pellucidum. From a fetal imaging perspective, holoprosencephaly is an important differential consideration when the CSP is absent on fetal US [41]. Fetal MRI can be employed in these patients to confirm the diagnosis and to establish the degree of brain involvement, which can be challenging on fetal ultrasound, particularly for the lobar and semilobar subtypes [42] (Figure 5).

### 4.2. Corpus Callosum Anomalies

Abnormalities of the corpus callosum can range from complete agenesis, hypogenesis or partial agenesis, or hypoplasia of the corpus callosum [43]. The corpus callosum cannot always be reliably evaluated on fetal ultrasound. However, abnormalities of the corpus callosum may be suspected based on absence of the CSP, dilation of the posterior horns of the lateral ventricles (also known as colpocephaly) or absence of the pericallosal artery on color Doppler [44]. Fetal MRI can not only aid in confirming the diagnosis of corpus callosum dysgenesis, but can also delineate the extent of corpus callosum abnormality as well as identify additional abnormalities in the brain [45]. The identification of additional associated anomalies portends a worse outcome which can be valuable for accurate prenatal counseling. In certain cases, the perinatal management may be affected as well. An example of this is the Sakoda complex, which is a rare condition described as a triad of abnormalities including agenesis of the corpus callosum, midline cleft lip and palate, and a basal encephalocele. This basal encephalocele cannot be reliably identified on prenatal US alone; however, if diagnosed on fetal MRI, it can be an indication for ex utero intrapartum treatment (EXIT) to airway procedure, as the cephalocele has the potential to obstruct the nasopharyngeal and oropharyngeal airways in the neonate [46] (Figure 6).

### 4.3. Hypoplastic Optic Nerves vs. Isolated Septal Deficiency

Hypoplastic optic nerve syndrome (also known as septo-optic dysplasia), is the primary differential consideration for isolated absence of the CSP. The term “septo-optic dysplasia,” also known as “de Morsier syndrome”, was first described by de Morsier in 1956 in postmortem cases of patients with optic nerve hypoplasia and agenesis of the septum pellucidum [47,48]. Many currently use the term “hypoplastic optic nerve syndrome”, given the diversity of findings in the brain related to hypoplastic optic nerves; however, the term “septo-optic dysplasia” is still frequently referenced. The postnatal MRI findings of hypoplastic optic nerve syndrome are unilateral or bilateral optic nerve hypoplasia, with or without one of the following additional findings: partial or complete absence of the septum pellucidum, dysgenesis of the corpus callosum, anomalies of the hypothalamic-pituitary axis, and malformations of cortical development, schizencephaly being most common [49]. While hypoplasia of the optic nerves themselves cannot be reliably assessed on fetal MRI, and is best evaluated by postnatal ophthalmology exam, abnormalities of the corpus callosum and neuronal migrational anomalies can be assessed. It is important to raise the possibility of hypoplastic optic nerve syndrome on fetal imaging so that neonatal ophthalmologic examination and endocrine screening studies can be pursued in order to prevent prolonged hypothalamic–pituitary dysfunction in these patients [50]. However, one must keep in mind that isolated septal deficiency in association with normal clinical outcomes is now more commonly encountered than previously thought, though it is a diagnosis of exclusion [51] (Figure 7).

## 5. Posterior Fossa Malformations

One of the most common reasons for referral for fetal MRI is to “rule out Dandy–Walker Malformation”. While classic Dandy–Walker malformation is a posterior fossa malformation encountered on prenatal ultrasound, there is a wide spectrum of abnormalities described in the Dandy–Walker continuum that can be classified on fetal imaging. Abnormalities of the cerebellar hemispheres and brainstem can also account for posterior fossa abnormalities identified on fetal ultrasound, which are better characterized on fetal MRI, and will also be discussed in this section.

### 5.1. Dandy-Walker Continuum

Suspected “Dandy–Walker Malformation” is a common reason for referral for fetal MRI, and is often a differential consideration for any posterior fossa abnormality identified on fetal US. Though the “Dandy–Walker Malformation” terminology is controversial, as there is ongoing inconsistency of its description in the literature, fetal MRI can be helpful in accurately characterizing abnormalities of the vermis and diagnosing classic Dandy–Walker Malformation, for which at our institution this diagnosis typically results in CSF diversion in the neonatal period [52]. Classic Dandy–Walker malformation has been described as having three imaging criteria: partial or complete absence of the vermis, cystic enlargement of the fourth ventricle, and an enlarged posterior fossa marked by upward displacement of the tentorium and torcular, seen as “torcular-lambdoid inversion” in the postnatal period [53]. Descriptions of vermis size by 2D measurements and vermis rotation, marked by the tegmento-vermian angle, which is measured between the brainstem and vermis, can be used to describe patients along the continuum on fetal MRI [54]. On fetal MRI a tegmento-vermian angle of ≥80° can potentially be used to aid in the diagnosis of classic Dandy–Walker malformation [55]. Vermis rotation is thought to be secondary to the abnormal persistence Blake pouch, which is a normal embryologic structure involved in 4th ventricular development that typically regresses by the end of the first trimester. Blake pouch remnant, which has been characterized by a tegmento-vermian angle of ≥18° in the ultrasound literature, can be seen in isolation or in the presence of vermian hypoplasia [56]. Fetal MRI can help distinguish between the two entities which is important as patients with vermian hypoplasia with co-existing Blake pouch remnant are more likely to have underlying genetic conditions than those with isolated Blake pouch remnant, which tend to have good outcomes but do need close postnatal follow-up to monitor for developing obstructive hydrocephalus [57] (Figure 8).

### 5.2. Other Abnormalities of the Cerebellar Hemispheres

Abnormalities of the cerebellar hemispheres comprise a wide spectrum of pathologies. Cerebellar hypoplasia may be suspected on fetal ultrasound based on a decreased transverse cerebellar diameter. Unilateral abnormalities of the cerebellar hemispheres can be difficult to assess on ultrasound due to technical factors. Global cerebellar hypoplasia is often seen in association with supratentorial malformations of cortical development, global cerebral brain injury or genetic syndromes [58]. Unilateral cerebellar hypoplasia can also potentially be seen in genetic abnormalities such as PHACES syndrome and COL4A1 mutations, though is more frequently seen in isolation or as a result of prior injury. Fetal MRI can identify cerebellar clefts or cerebellar hemorrhage with greater sensitivity and specificity than fetal US [59]. This can provide important information for prenatal counseling as cerebellar hemorrhage can potentially have postnatal long term neurodevelopmental consequences [60] (Figure 9).

### 5.3. Brainstem Anomalies

Abnormalities of the brainstem are not well delineated on fetal US, and in these cases fetal MRI can play a key role in delineating brainstem morphology and narrowing the differential diagnosis. One example is Joubert syndrome and related disorders (JSRD), which comprise a genetically heterogeneous group of diseases, most resulting from mutations to genes encoding ciliary proteins. While fetal ultrasound may suggest a vermis abnormality, a hypoplastic vermis without a co-existing Blake pouch remnant is easier to define on fetal MRI and provides an important clue to look for the classic brainstem “molar tooth sign” which is characteristic of the disease [61,62] (Figure 10). A kinked or “z-shaped” brainstem is also a finding that is much easier to delineate on fetal MRI than fetal ultrasound, which can signify an underlying dystroglycanopathy (Walker–Warburg phenotype), L1CAM mutation or tubulinopathy [63]. These diagnostic considerations typically have poor prognosis, with profound implications for prenatal counseling, highlighting the value of fetal MRI in these cases.

## 6. Conclusions

Fetal MRI is a valuable and currently widely accessible tool that can help to narrow the differential and in many cases aid in establishing specific diagnoses in suspected brain abnormalities in the fetus. This has greatly improved the accuracy of counseling so that expecting parents can make important decisions prenatally and have the opportunity to prepare for the expected postnatal course. In the era of fetal surgery, fetal MRI plays a key role in identifying the best candidates for fetal surgery and providing better anatomic information necessary for surgical planning. Finally, fetal MRI can aid in providing information that will guide delivery planning, highlighting the need for neonatal interventions and subspecialty care, such that the best treatment can be made available at the time of delivery. The role of fetal MRI continues to evolve in maternal–fetal medicine, neurology, neurosurgery, and neonatology clinical practices and holds promise to aid in improved outcomes for these patients in the future.

## Figures and Tables

**Figure 1 diagnostics-12-00764-f001:**
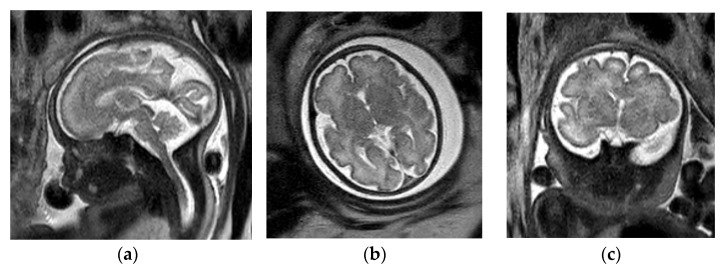
Fetal brain MRI protocol. Normal brain MRI in a 32-week, 6-day GA fetus. Sagittal (**a**), axial (**b**) and coronal (**c**) T2-SSFSE images in brain are used to evaluate brain parenchyma signal and morphology. Axial DWI (**d**) can be used to look for cerebral infarction. Axial EPI (**e**) allows for evaluation of intracranial hemorrhage with a greater sensitivity. Axial T1-weighted image (**f**) has limited spatial resolution however can be helpful to evaluate for intracranial lipoma, hemorrhage or thrombosis.

**Figure 2 diagnostics-12-00764-f002:**
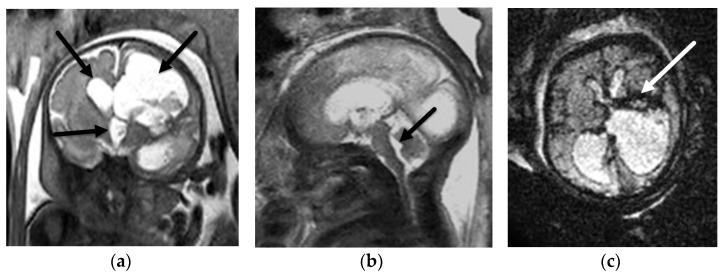
Aqueductal stenosis. Coronal (**a**) and sagittal (**b**) T2-SSFSE and axial EPI (**c**) images from a 34-week, 2-day GA fetus referred for ventriculomegaly on fetal ultrasound. There is severe ventriculomegaly of the lateral and third ventricles ((**a**), black arrows) with a normal sized 4th ventricle ((**b**), black arrow) consistent with aqueductal stenosis. The etiology, not identified on fetal US, is a left grade 4 germinal matrix hemorrhage best seen on the EPI blood sequence ((**c**), white arrow) with hemosiderin staining along the ependymal margins of the ventricular system. Associated porencephaly of the left cerebral hemisphere is best seen on the coronal (**a**) image.

**Figure 3 diagnostics-12-00764-f003:**
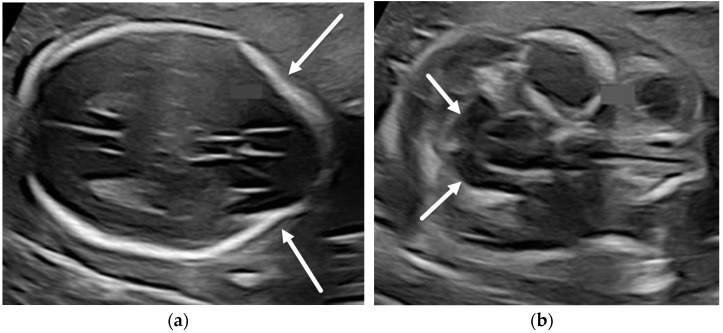
Chiari II malformation. Fetal ultrasound of a 19-week, 6-day GA fetus with myelomeningocele (not imaged) demonstrating the “lemon sign” ((**a**), arrows) and the “banana sign” ((**b**), arrows). Axial (**c**) and sagittal (**d**) T2-SSFSE images in the same patient show effacement of the extra-axial CSF spaces (**c**) and severe hindbrain herniation ((**d**), arrow) consistent with Chiari II malformation.

**Figure 4 diagnostics-12-00764-f004:**
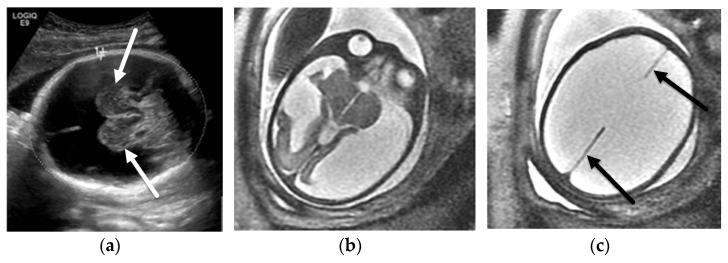
Hydranencephaly. A 35-week, 6-day GA fetus referred for ventriculomegaly on screening ultrasound. Axial ultrasound image (**a**) through the head demonstrates near complete absence of the cerebral hemispheres with relative sparing of the thalami (arrows). Axial T2-SSFSE images through the fetal brain (**a**,**b**) confirm fetal ultrasound findings with near complete absence of the cerebral hemispheres with relative sparing of the thalami and temporo-occipital lobes consistent with hydranencephaly. There is perseveration of the falx cerebri ((**c**), arrows), which is characteristic of this diagnosis.

**Figure 5 diagnostics-12-00764-f005:**
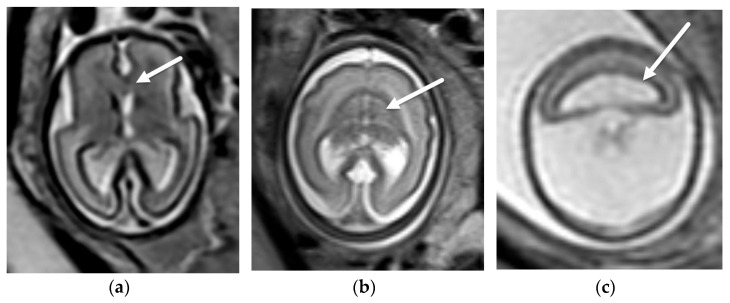
Holoprosencephaly spectrum. Axial T2-SSFSE images of the brain in fetuses with holoprosencephaly spectrum. Fetal MRI of a fetus with lobar holoprosencephaly at 24 weeks and 2 days GA (**a**) demonstrates near complete separation of the cerebral hemispheres with a small area of incomplete separation (arrow) in the frontal lobes. Fetal MRI of a fetus with semi-lobar holoprosencephaly at 24 weeks and 2 days GA (**b**) demonstrates absence of normal separation of the frontal lobes, basal ganglia and thalami (arrow). Fetal MRI of a 21-week GA fetus with alobar holoprosencephaly (**c**) demonstrates complete absence of normal separation of the cerebral hemispheres with a monoventricle (arrow).

**Figure 6 diagnostics-12-00764-f006:**
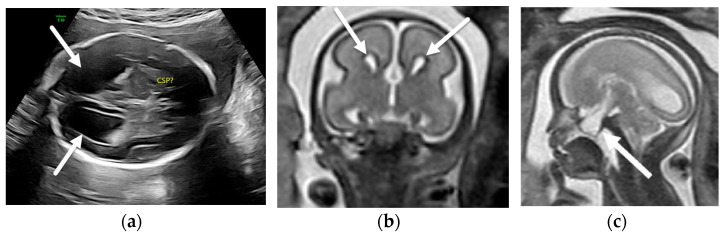
Agenesis of the corpus callosum in the Sakoda complex. Axial image of fetal ultrasound at 24 weeks GA (**a**) for cleft lip and palate (not shown) demonstrates absence of the CSP and colpocephaly (arrows) suspicious for agenesis of the corpus callosum. This is confirmed on subsequent fetal MRI at 24 weeks and 1 day GA. Coronal T2-SSFSE image of the brain (**b**) demonstrates upturned frontal horns of the lateral ventricles (arrows) and absence of the corpus callosum connecting the cerebral hemispheres. (**c**) Sagittal T2-SSFSE image in the same fetus reveals a large sphenoethmoidal encephalocele (arrow) projecting into the nasopharynx.

**Figure 7 diagnostics-12-00764-f007:**
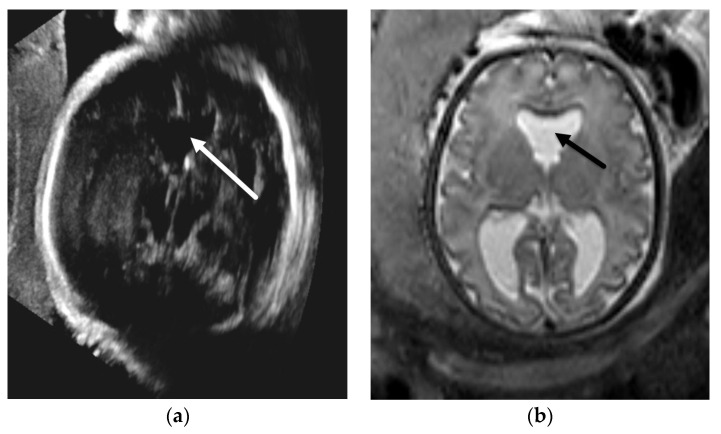
Isolated septal deficiency. Axial ultrasound image of the fetal brain in a 33-week GA fetus (**a**) demonstrating absence of the CSP (arrow). Axial T2-SSFSE image of the brain in the same patient (**b**) confirms absence of the CSP (arrow) with associated mild ventriculomegaly of the lateral ventricles. This patient had postnatal follow-up with normal ophthalmology exam and normal endocrine work up. Postnatal brain MRI did not demonstrate any additional anomalies.

**Figure 8 diagnostics-12-00764-f008:**
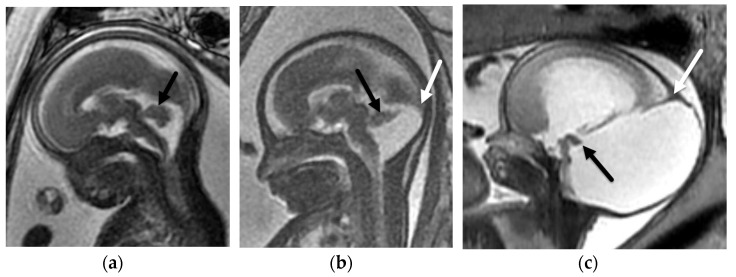
Dandy–Walker Continuum. Sagittal BTFE/FIESTA image from fetal MRI at 25 weeks and 3 days GA with isolated Blake pouch remnant. (**a**) The vermis (arrow) is normal in size and morphology; however, it is mildly rotated. Sagittal BTFE/FIESTA image from fetal MRI at 27 weeks GA with vermian hypoplasia with co-existing Blake pouch remnant (**b**) demonstrates a hypoplastic and rotated vermis (black arrow); however, the torcular (white arrow) is not elevated. Sagittal T2-SSFSE image from fetal MRI of classic Dandy–Walker malformation (**c**) at 26 weeks and 6 days GA with non-visualization of the vermis posterior to the tectum (black arrow) and a markedly elevated torcular (white arrow) with enlargement of the posterior cranial fossa and obstructive hydrocephalus.

**Figure 9 diagnostics-12-00764-f009:**
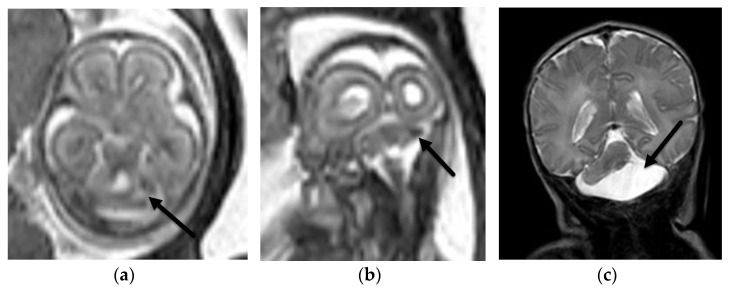
Unilateral cerebellar injury. Axial (**a**) and coronal (**b**) T2-SSFSE images from fetal MRI performed at 20 weeks and 6 days GA demonstrates a cleft in the left cerebellar hemisphere with T2 hypointense blood products (arrows). (**c**) Coronal T2-FSE image from postnatal MRI at day of life 1 in the same patient demonstrates absence of the left cerebellar hemisphere (arrow) compatible with evolution of prenatally diagnosed cerebellar injury.

**Figure 10 diagnostics-12-00764-f010:**
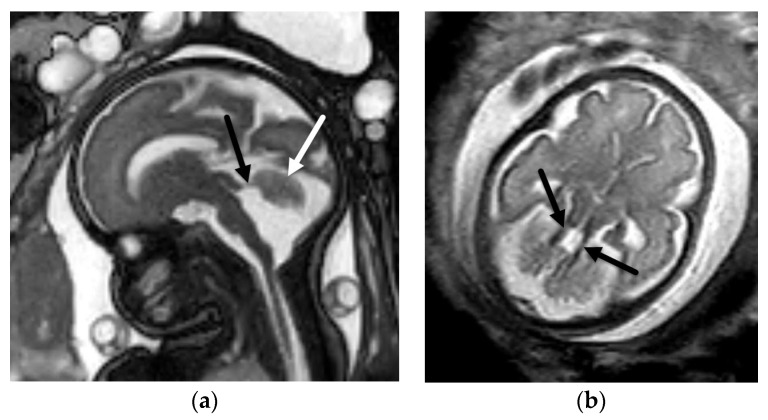
Joubert syndrome. Sagittal BTFE/FIESTA image (**a**) in a 29-week, 6-day GA fetus demonstrates vermian hypoplasia (white arrow) with horizontal orientation of the superior cerebellar peduncles (black arrow). Axial T2-SSFSE image (**b**) in the same fetus confirms the “molar tooth” sign with thickened and horizontal orientation of the superior cerebellar peduncles (arrows). Postnatal genetic testing confirmed Joubert syndrome in this patient.

## Data Availability

Not applicable.

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
