# Peer review of "Clinical Applications of Fetal MRI in the Brain"

_diagnostics, 2022, doi:10.3390/diagnostics12030764_

Round 1

Reviewer 1 Report

Fetal MRI: which fieldstrength do the authors recommend? There are also ISUOG Guidelines for fetal MRI protocols that may be mentioned (UOG 2017).

Obstructive Hydrocephalus: are there morphological signs that allow to recognize increased intracranial pressure as a consequence of the obstructiuon of CSF spaces?

Does it require volumetry to diagnose decreased cerebral volume?

Does normal appearance of the optic structures exclude postnatal impairment of visual function?

The term "Dandy Walker Continuum" should not be used any more: 

Guibaud l. et al, Prenatal Diagnosis 2012:32:185-93

Reviewer 2 Report

Dear Author

I read the manuscript titled " Clinical Applications of Fetal MRI in the Brain " with interest and found it extremely informative, well-organized and well-written.

The content is very coherent, starting from the routine ultrasound anatomical survey and organizing the findings based on the three main abnormalities seen in ultrasound. While the list is by no means exclusive, it covers the major categories.

The images are informative and the captions provide just the right amount of explanation.

While there is much more to be said about the limitations of fetal MRI, the limitations are also properly mentioned.

I find it an informative, easy and pleasurable read.
